# Exploring the Roles of Vitamins C and D and Etifoxine in Combination with Citalopram in Depression/Anxiety Model: A Focus on ICAM-1, SIRT1 and Nitric Oxide

**DOI:** 10.3390/ijms25041960

**Published:** 2024-02-06

**Authors:** Omar Gammoh, Aseel Ibrahim, Ala Yehya, Abdelrahim Alqudah, Esam Qnais, Sara Altaber, Osama Abo Alrob, Alaa A. A. Aljabali, Murtaza M. Tambuwala

**Affiliations:** 1Department of Clinical Pharmacy and Pharmacy, Faculty of Pharmacy, Yarmouk University, Irbid 21163, Jordan; alaa.yehya@yu.edu.jo (A.Y.); osama.yousef@yu.edu.jo (O.A.A.); 2Faculty of Sciences, Yarmouk University, Irbid 21163, Jordan; aseel.mane7@yahoo.com; 3Department of Clinical Pharmacy and Pharmacy Practice, Faculty of Pharmaceutical Sciences, The Hashemite University, Zarqa 13133, Jordan; abdelrahim@hu.edu.jo; 4Department of Biology and Biotechnology, Faculty of Science, The Hashemite University, Zarqa 13133, Jordan; esamqn@hu.edu.jo (E.Q.); sara.taber12@gmail.com (S.A.); 5Department of Pharmaceutics and Pharmaceutical Technology, Yarmouk University, Irbid 21163, Jordan; alaaj@yu.edu.jo; 6Lincoln Medical School, Brayford Pool Campus, University of Lincoln, Lincoln LN6 7TS, UK; mtambuwala@lincoln.ac.uk

**Keywords:** citalopram, ICAM-1, SIRT1, depression

## Abstract

The study of intercellular adhesion molecule-1 (ICAM-1) and SIRT1, a member of the sirtuin family with nitric oxide (NO), is emerging in depression and anxiety. As with all antidepressants, the efficacy is delayed and inconsistent. Ascorbic acid (AA) and vitamin D (D) showed antidepressant properties, while etifoxine (Etx), a GABAA agonist, alleviates anxiety symptoms. The present study aimed to investigate the potential augmentation of citalopram using AA, D and Etx and related the antidepressant effect to brain and serum ICAM-1, SIRT1 and NO in an animal model. BALB/c mice were divided into naive, control, citalopram, citalopram + etx, citalopram + AA, citalopram + D and citalopram + etx + AA + D for 7 days. On the 8th day, the mice were restrained for 8 h, followed by a forced swim test and marble burying test before scarification. Whole-brain and serum expression of ICAM-1, Sirt1 and NO were determined. Citalopram’s antidepressant and sedative effects were potentiated by ascorbic acid, vitamin D and etifoxine alone and in combination (*p* < 0.05), as shown by the decreased floating time and rearing frequency. Brain NO increased significantly (*p* < 0.05) in depression and anxiety and was associated with an ICAM-1 increase versus naive (*p* < 0.05) and a Sirt1 decrease (*p* < 0.05) versus naive. Both ICAM-1 and Sirt1 were modulated by antidepressants through a non-NO-dependent pathway. Serum NO expression was unrelated to serum ICAM-1 and Sirt1. Brain ICAM-1, Sirt1 and NO are implicated in depression and are modulated by antidepressants.

## 1. Introduction

Psychological stress is often associated and comorbid with depression and anxiety disorders [1]. Major depressive disorder (MDD) and anxiety are the leading causes of disability worldwide [2]. Growing lines of literature refer to the implication of neuroinflammation in MDD and anxiety [3,4]. The relationship between psychiatric disorders and neuroinflammation is well established. Neuroinflammation refers to the combined immune response and inflammation within the CNS. Increasing numbers of publications in the literature refer to the implication of neuroinflammation in depressive disorder [3,4].

Upon activation, the majority of microglia become amoeboid and induce the production of proinflammatory cytokines (TNF-α, IL-1β and IL-6) [5], which in turn increase the levels of the presynaptic transporters of monoamines, thus leading to their depletion from the synaptic space. Additionally, cytokines activate the indoleamine 2,3 dioxygenase (IDO) enzyme, which breaks down tryptophan, the precursor of serotonin, to yield kynurenine, which is converted into quinolinic acid and, subsequently, into glutamate [6,7]. Furthermore, inflammatory processes and altered leukocyte trafficking play a role in the disruption of the blood–brain barrier in psychiatric disorders, supporting the hypothesis of the involvement of the central nervous system and peripheral tissues in ongoing inflammatory processes [8]. 

Adhesion molecules play an important role in leukocyte recruitment and their role in psychiatry is emerging. Intercellular adhesion molecule-1 (ICAM-1), a marker of inflammation, is an immunoglobulin (Ig)-like transmembrane glycoprotein that is overexpressed in the endothelial lumen. ICAM-1 facilitates leukocyte transmigration across the endothelium of many cell types and plays a key role in the function of the blood–brain barrier (BBB), which helps in the migration of molecules to and from the brain [4,9]. The membrane-bound ICAM-1 is primarily expressed in several parts of the central nervous system (CNS) [10].

A soluble form of the molecule, the soluble intercellular adhesion molecule-1 (sICAM-1), is found in serum and is considered a peripheral inflammatory marker associated with several inflammatory states [11]. While little evidence exists about the implications of membrane-bound ICAM-1 in the CNS, the blood levels of sICAM-1 were found to be increased in patients with depression [12]. 

Sirtuins are a family of NAD+-dependent deacetylases that have a crucial role in anti-oxidative and anti-inflammatory balance and homeostasis [13]. SIRT1 is a member of the sirtuin family. The role of Sirt1 in depression is emerging. Two studies showed lower Sirt1 blood levels in patients with depression versus controls [14,15]. Moreover, Sirt1 activation in animal models of depression was associated with antidepressant effects [16,17]. Nitric oxide (NO) is produced from L-arginine by the enzymatic conversion of the enzyme NO synthase (NOS) [18] and is highly implicated in depression and anxiety [19,20]. Moreover, the expression of ICAM-1 and Sirt1 is thought to be, in part, dependent on NO pathways. Nitric oxide increases the expression of ICAM-1 in epithelial cells [21,22]. In addition, the inhibition of the NO synthesis reduced the levels of SIRT1 protein levels [23]. 

Although selective serotonin reuptake inhibitors (SSRIs) are a first-line therapy for MDD, their delayed onset and the inconsistency of antidepressant efficacy are major limitations [24]. While the need for augmenting antidepressants is critical, combining traditional antidepressants from the same or different families is not recommended due to serious adverse reactions such as asserotoni symptoms, a higher risk of weight gain and drug interactions [25].

There is emerging evidence linking depression with vitamin D. The risk of depression may be further exacerbated by low serum levels of vitamin D [26]. Some evidence indicates that vitamin D could play a role as a neuroactive molecule that enhances the expression of neurotransmitters [27]. Ascorbic acid, or vitamin C, is a known antioxidant and water-soluble vitamin that is involved in many cellular processes. In one preclinical study, ascorbic acid exerted an antidepressant effect in mice. In addition, it potentiated the effects of fluoxetine and imipramine [28]. The results of another pilot study suggest that vitamin C may be an effective adjuvant agent in the treatment of MDD in paediatric patients receiving SSRIs [29]. Etifoxine is a GABAA agonist and has an anxiolytic role that could be a potential adjuvant to depression and anxiety [30,31]. 

The acute restraint model involves the acute immobilisation of animals for a period of time between 6 and 8 h. This stress-induced model demonstrates anxiety and behaviour-like behaviour in animals, a model that is associated with changes in biochemical markers [8,32]. Animal models using BALB/c mice showed either no response or a very poor response to subchronic treatment with antidepressants such as citalopram. For example, the subchronic administration of citalopram failed to decrease floating immobility time in the forced swim test in BALB/c mice [33,34].

To the best of our knowledge, scattered evidence points to the possible effect of ascorbic acid and vitamin D in alleviating depressive symptoms. Furthermore, etifoxine showed non-negligible efficacy in anxiety compared to benzodiazepines. No previous studies have examined the potential adjuvant effects of ascorbic acid, vitamin D and etifoxine on the antidepressant and anxiolytic effects of citalopram with the expression of ICAM-1, Sirt1 and NO in the brain and serum of BALB/c mice. Therefore, the present study had two objectives: (1) to investigate the efficacy of ascorbic acid, vitamin D and etifoxine in potentiating the efficacy of citalopram and (2) to relate the antidepressant effect to the expressions of ICAM-1, Sirt1 and NO in the brain and serum of BALB/c mice.

## 2. Results

### 2.1. Behavioural Tests

According to the forced swim test, there was a significant increase in the floating time in the naive group versus the control group (*p* < 0.05). The citalopram-treated group demonstrated a significantly longer floating time compared to the naive group (*p* < 0.05). All the adjuvants significantly reduced the floating time (*p* < 0.05) compared to citalopram. According to the marble burying test, the number of marbles buried by the control group was significantly higher compared to the naive group (*p* = 0.003), compared to the control group, where the citalopram-treated group buried significantly fewer marbles (*p* = 0.009). The frequency of ambulation of the open field test of the groups treated with (cita + etx, cita + AA) and (cita + etx + AA + D) showed a significant reduction (*p* < 0.05) in ambulation frequency compared to citalopram. Furthermore, the results of the rearing frequency showed that the control demonstrated a significantly higher (*p* < 0.05) rearing frequency compared to the naive group and that all the combinations (cit + etx, cita + AA, cit + D and cit + AA + D) demonstrated a significant decrease (*p* < 0.05) in the rearing frequency compared to citalopram. None of the adjuvants showed significant improvements compared to the citalopram group (*p* > 0.05) (Figure 1).

### 2.2. ICAM-1 and SIRT1

ICAM-1 expression in the brain was significantly higher in the control group compared to the naive group (*p* = 0.005). Citalopram did not affect ICAM-1 expression; however, the Cita + C + D + Etx group had a significantly lower ICAM-1 level (*p* = 0.01) than the control group. The brain Sirt1 expression of the control group was significantly lower (*p* = 0.02) compared to the naive group. Citalopram did not increase the Sirt1 levels; however, the Cita + C + D + Etx group had significantly higher Sirt1 expression (*p* = 0.02) than the control group. The serum expression of ICAM-1 and Sirt1 did not vary between the groups (*p* > 0.05) (Figure 2).

### 2.3. Nitric Oxide

The control group demonstrated significantly higher NO brain levels compared to the naive group (*p* < 0.05). Only Cit + C showed a reduction in the brain NO levels compared to the control (*p* < 0.05). The serum NO levels were significantly higher in the control group compared to the naive group (*p <* 0.05). Furthermore, all the treated groups had lower serum NO levels than the control group (*p* < 0.05) (Figure 3).

## 3. Discussion

The present study aimed to investigate neuroinflammation in psychiatric disorders. We investigated the role of vitamins C and D and etifoxine as potential adjuvants to citalopram in a B/C model of anxiety and depression. In our model, citalopram did not exert an antidepressant effect as the floating time did not decrease significantly compared to the control group; however, etifoxine, ascorbic acid and vitamin D alone and in combination were able to enhance the antidepressant effect of citalopram. The anxiolytic effect of citalopram was evaluated mainly using the marble burying test. The citalopram-treated group showed a significantly lower number of marbles buried, indicating significant anxiolytic efficacy versus the control group; however, this anxiolytic efficacy was not further enhanced by ascorbic acid, vitamin D, etifoxine, or their combination compared to citalopram alone. Furthermore, the open field test used to screen for anxiolytic and sedative effects revealed that the sedative effect of citalopram was not evident in the treated mice; however, the sedative effect was evident with citalopram in combination with etifoxine, ascorbic acid, vitamin D alone and then combined.

In our study, we used ascorbic acid, vitamin D and etifoxine as potential adjuvants due to their established antioxidant and anti-inflammatory properties and their emerging antidepressant properties. Citalopram, as with other SSRIs, inhibits the serotonin carriers, therefore increasing the serotonin (a monoamine) levels at the synaptic cleft. Although it is a first-choice drug for anxiety and depression due to its safety and minimal drug interactions, about 50% of patients experience inadequate responses to SSRIs in the best conditions [35,36]. Furthermore, reviewing three decades of randomised controlled trials of antidepressants revealed that the antidepressant response rate is approximately 54% compared to a placebo response rate of 37% [37]. Therefore, the monoamine-based mechanism does not result in the optimal resolution of the patient’s symptoms. Furthermore, the use of two antidepressants could lead to detrimental effects such as serotonin syndrome, weight gain and sexual dysfunction [25]. As a result, enhancing the efficacy of SSRIs should be carried out through other strategies. In light of the literature, several potential natural and synthetic antioxidant and anti-inflammatory molecules could play an adjuvant antidepressant role. For example, vitamin D is believed to act as a neuroactive steroid [27,38] that plays a key role in the expression of neurotransmitters, the synthesis of antioxidants and neurotropic factors, thus providing a rationale for its potential antidepressant role. Although its role in depression is now still emerging, it is evident that low vitamin D levels are associated with depression [26].

Ascorbic acid, or vitamin C, is known for its antioxidant, anti-inflammatory and other metabolic effects, which could explain its adjuvant effect with antidepressants [29]. In a preclinical study, ascorbic acid exerted an antidepressant effect in mice. In addition, it potentiated the effects of fluoxetine and imipramine [28]. Etifoxine, a new GABAA agnostic, is believed to be a non-benzodiazepine anxiolytic with the advantage of being non-addictive [30]. Benzodiazepines such as lorazepam and bromazepam, are often co-prescribed with SSRIs to ‘bridge’ delayed SSRI effects and alleviate their symptoms of jitteriness. This is based on the immediate actions resulting from GABAA receptor activation, resulting in hyperpolarisation, which subsequently leads to a calming and anxiolytic effect. Despite the scarcity of data, etifoxine has been demonstrated to be non-inferior to benzodiazepines [39]. Also, etifoxine has never been tested as a benzodiazepine alternative along with SSRI for depression and anxiety. Moreover, the antidepressant activity of etifoxine has not been previously studied. The sedative effect could be mainly attributed to etifoxine; however, the antidepressant effects are related to the anti-inflammatory and associated mechanisms [26,30,40]. In addition, the study highlighted the implications of the central and peripheral expression of ICAM-1, SIRT1 and NO in psychiatric distress in accordance with previous research [41,42]. The acute immobility stress increased brain ICAM-1, caused a decrease in brain Sirt1 expression and increased the NO levels. The antidepressant effect of the combination (Cit + AA + D + etx) normalised both the ICAM-1 and SIRT1 brain levels but was independent of NO. The serum NO levels of NO were increased due to the acute immobility test and decreased with antidepressant treatments, which are not related to ICAM-1 and Sirt1 expression. Depression and neuroinflammation are tightly related [36]. The role of nitric oxide is established in depression. The bulk of the evidence supports an increase in NO during depression. For example, NO and its metabolites were higher in suicide attempters with respect to the controls [19,20]. Moreover, studies pointed out elevated NO levels in MDD patients, and, interestingly, NO levels were normalised after antidepressant therapy [43]. In support of the potential role in depression, a growing body of evidence has demonstrated that some antidepressants exert a NO-lowering effect. A study revealed that L-arginine antagonised the effects of the classic tricyclic antidepressant imipramine [44]. Several medications have been shown to exert antidepressant effects through NOS inhibition, including tramadol, bupropion, venlafaxine, lithium and ketamine [45,46]. Furthermore, many new investigational amino acid NOS modifiers have demonstrated interesting results, such as L-NG-nitroarginine, L-NG-monomethyl arginine and NG-propyl-L-arginine [47]. The role of ICAM-1 and Sirt1 is still emerging. Our findings showed an increase in brain ICAM-1 and a decrease in Sirt1 in the depressed group; this modulation was associated with an increase in NO. In agreement with our findings, previous studies demonstrated that ICAM-1 is upregulated via an NO-dependent mechanism, most probably via gene upregulation mechanisms involving transcriptional factors [21]. Similarly, inhibiting nitric oxide synthesis resulted in a reduction in Sirt1 protein expression [23]. The brain, but not the soluble ICAM-1 and Sirt1, was modulated, according to our findings. Moreover, the changes in the serum NO levels did not match those of ICAM-1 and Sirt1. Soluble ICAM-1 was increased in depression and in chronic inflammatory conditions, as reviewed in [4]. Our findings can be explained by the acute period of stress applied. The present study adds to the existing literature in several ways. The study tries to potentialise the effect of citalopram using a BALB/c mouse model known for its resistance to the subchronic administration of SSRI. The study recruited different safe and available ‘out of the box” adjuvants that work through a multi-mechanistic approach. Our findings will pave the road to further investigations aimed ultimately at augmenting the efficacy and fastening the onset of the action of antidepressants, thereby optimising the patient’s experience and outcomes with minimal side effects. Furthermore, our study highlighted the role of relatively new markers for neuroinflammation by focusing on the roles of ICAM-1, SIRT1 and NO in psychiatric disorders, namely stress, anxiety and depression. In addition, the study examined the expression of markers in both the central nervous system and the serum in an attempt to study the behaviour and potential effects of inflammation markers outside the central nervous system. 

On the other hand, some aspects of this study can be improved in future studies. For example, the acute stress model applied to mice (8 h) can be further enhanced using chronic repeated stress; other models could also be used as models of inflammatory depression and anxiety, such as the lipopolysaccharide model. Furthermore, other antidepressants, including serotonin–norepinephrine reuptake inhibitors (SNRIs) or tricyclic antidepressants (TCAs), could be used with different durations of treatment. In addition, other adjuvants could be evaluated for their antidepressant potential.

## 4. Materials and Methods

### 4.1. Animal Studies

Male BALB/c mice (from the Animal House Facility of Yarmouk University, Irbid, Jordan) were used in the present study. The mice were 6–8 weeks old and weighed 25 g. The BALB/c line was selected due to its suitability for achieving the study objective of enhancing citalopram efficacy. The mice were kept in separate cages at a temperature of 25 °C with 50–60% humidity and continuous air ventilation. The research was carried out according to the international ethical standards for the care and use of laboratory animals, and the IRB committee of study was approved by Yarmouk University and the Dean of Scientific Research Project Number (51/2022).

### 4.2. Study Design and Treatments

After 2 days of habituation, mice were randomly allocated to the following groups (n = 6–10 per group): naive (unstressed), control (stressed), citalopram (10 mg/kg/day), citalopram + etifoxine (50 mg/kg/day), citalopram + ascorbic acid (10 mg/kg/day), citalopram + vitamin D (1200 IU/kg/week) and citalopram + ascorbic acid + vitamin D + etifoxine for 7 days, followed by acute immobilisation for 8 h on the 8th day. All the doses were chosen based on the relevant literature [33,48,49,50,51,52]. All treatments were purchased from (Santa Cruz, Santa Cruz, CA USA).

### 4.3. Acute Restraint Model

To induce anxiety in the mice, an acute immobility stress test was performed according to the method of Machawal and Kumar [32] with slight modifications. The mice were restrained for 8 h while breathing allowed.

### 4.4. Behavioural Paradigms

#### 4.4.1. Forced Swim Test

The FST is the most commonly used behavioural model for screening antidepressant-like activity in rodents [53]. The mice were individually forced to swim for 5 min in an open glass chamber (25 × 15 × 25 cm^3^) containing fresh water to a height of 15 cm and maintained at 26 ± 1 °C. The floating time (FT) was defined as the time at which the mice stopped moving completely while in the water.

#### 4.4.2. Marble Burying Test

A cage (17.5 × 10 × 5.5 inches) was filled to a depth of approximately 5 cm with husk bedding material that was evenly distributed on a flat surface across the entire cage. Twenty glass marbles (1.4 cm in diameter) were then evenly spaced in a 4 × 5 grid on the surface of the bedding. During the testing phase, each mouse was placed in the cage and allowed to explore it for 30 min. At the end of the test, the mice were removed from the cage and the number of marbles buried with bedding up to 2/3 of their depth was counted as reported in [54].

#### 4.4.3. Open Field Test

An OFT was performed to assess locomotion, anxiety and sedation in the mice, as in [55]. Briefly, the mice were placed in a central square and allowed to move freely for 5 min. The field was located in a test room and was lit by indirect lighting. The procedure was performed in an empty room to minimise noise and distractions. The open field maze was cleaned between each mouse using 70% ethyl alcohol. The locomotion activity (represented by the number of lines crossed) and sedation (represented by the rearing frequency) were recorded.

### 4.5. Tissue Collection

After performing the behavioural tests, the mice were sacrificed, blood was collected and serum was obtained using standard protocols. Also, whole brains were collected and stored at −80 for further analysis.

#### 4.5.1. Western Blots

Western blot was performed according to standard protocols as in [56] Mirzaii-Dizgah et al. (2020) with some modifications. The total protein was quantified using a bicinchoninic acid assay kit (Bioquochem, Oviedo, Spain), and an equal amount of protein was separated using a sodium dodecyl sulphate-polyacrylamide gel and then the proteins were transferred to a nitrocellulose membrane (Thermo Fisher Scientific, Waltham, MA, USA). The membrane was then blocked for 1 h at room temperature using 3% bovine serum albumin (BSA) before incubating it overnight with either ICAM-1 or SIRT1 primary antibodies (Abcam, Cambridge, UK). The membrane was washed three times with washing buffer (Tween-20/Tris-buffered saline) before incubating it with the respective secondary antibody (Mybiosource, San Diego, CA, USA) for 1 h at room temperature. Then, the membrane was washed three times before incubating it with ECL substrate (ThermoScientific, USA) for one minute and imaging it using the chemiLITE Chemiluminescence Imaging System (Cleaver Scientific, Rugby, UK). Equal gel loading was determined using actin as a housekeeping gene (Mybiosource, USA). The intensity of the bands was measured using Image J software (National Institutes of Health, USA, https://imagej.net/ij/docs/index.html, accessed on 12 September 2023).

#### 4.5.2. Nitric Oxide Assay

Blood was collected and serum was obtained using standard protocols. The accumulation of nitrate, an indicator of the production of NO, was determined using a colorimetric assay with a Griess reagent [57]. Serum nitrate was assayed using a Nitric Oxide Assay kit (Sunlong, Shanghai, China) according to the manufacturer’s instructions. 

### 4.6. Statistical Analysis

All the data obtained from the behavioural tests and ICAM-1, Sirt1 and NO were analysed using one-way ANOVA and a subsequent Tukey’s post hoc test. A significance threshold was established at *p* < 0.05. The data are presented as the mean plus the standard error of the mean.

## 5. Conclusions

Citalopram’s antidepressant and sedative effects are potentiated by ascorbic acid, vitamin D and etifoxine alone or in combination. The increase in NO expression in the brain NO increase was associated with an increase in ICAM-1 and a decrease in SIRT1, but only in the depression state. Antidepressants normalised the brain ICAM-1 and Sirt1 levels through a non-NO-dependent pathway. The serum levels of ICAM-1, SIRT1 and NO were not related to the psychiatric observations in mice. More studies are required to fully uncover the role of neuroinflammation in psychiatric disorders.

## Figures and Tables

**Figure 1 ijms-25-01960-f001:**
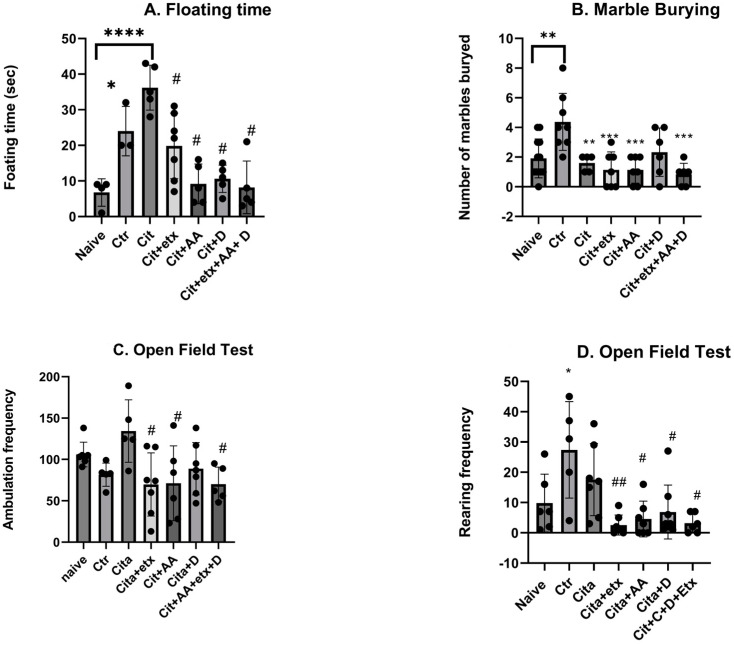
(**A**) floating time among the different groups. ANOVA followed by Tukey’s post hoc analysis. Values are expressed as the mean SEM (ANOVA followed by Tukey’s test). F(6, 27) = 12.48; *p* < 0.0001. * *p* < 0.05, **** *p* < 0.0001 vs. naive and # *p <* 0.00001 vs. citalopram. Ctr: control; Cit: citalopram; etx: etifoxine; D: vitamin D; AA: ascorbic acid; SEM, standard error of the mean. (**B**) The number of buried marbles in the group ANOVA followed by Tukey’s post hoc analysis. Values are expressed as the mean SEM (ANOVA followed by Tukey’s test). F(6, 44) = 6.29; *p* = 0.0001. ** *p* <0.05; *** *p* < 0.001 vs. control. Ctr: control Cit: citalopram; etx: etifoxine; D: vitamin D; AA: ascorbic acid; SEM, standard error of the mean. (**C**) The ambulation frequency per group ANOVA followed by Tukey’s post hoc analysis. Values are expressed as the mean SEM (ANOVA followed by Tukey’s test). F(6, 35) = 3.19; *p* = 0.001; # *p* < 0.05 vs. citalopram. Ctr: control Cit: citalopram; etx: etifoxine; D: vitamin D; AA: ascorbic acid; SEM, standard error of the mean. (**D**) The rearing frequency per group ANOVA followed by Tukey’s post hoc analysis. Values are expressed as the mean SEM (ANOVA followed by Tukey’s test). F(6, 35) = 3.19; *p* = 0.001. * *p* < 0.05 vs. naive; ## *p* < 0.001 vs. Ctr; # *p <* 0.05 vs. Ctr. Ctr: control Cit: citalopram; etx: etifoxine; D: vitamin D; AA: ascorbic acid; SEM, standard error of the mean.

**Figure 2 ijms-25-01960-f002:**
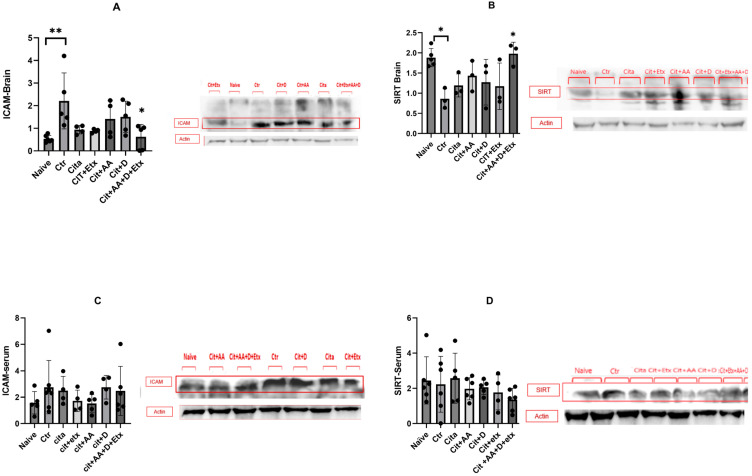
(**A**) Brain ICAM-1 expression among the different groups ANOVA followed by Tukey’s post hoc analysis. Values are expressed as the mean SEM (ANOVA followed by Tukey’s test). F(6, 16) = 4.02; *p* = 0.005. * *p <* 0.05; ** *p <* 0.005. Ctr: control; Cit: citalopram; Etx: etifoxine; D: vitamin D; AA: ascorbic acid; SEM, standard error of the mean. (**B**) Brain Sirt1 expression among the different groups ANOVA followed by Tukey’s post hoc analysis. Values are expressed as the mean SEM (ANOVA followed by Tukey’s test). F(6, 16) = 3.95; *p* = 0.01. * *p <* 0.05. Ctr: control; Cit: citalopram; etx: etifoxine; D: vitamin D; AA: ascorbic acid; SEM, standard error of the mean. (**C**) Serum ICAM-1 expression among the different groups ANOVA followed by Tukey’s post hoc analysis. Values are expressed as the mean SEM (ANOVA followed by Tukey’s test). F(6, 28) = 0.79; *p* = 0.58. Ctr: control; Cit: citalopram; Etx: etifoxine; D: vitamin D; AA: ascorbic acid; SEM, standard error of the mean. (**D**) Serum Sirt1 expression among the different groups ANOVA followed by Tukey’s post hoc analysis. Values are expressed as the mean SEM (ANOVA followed by Tukey’s test). F(6, 28) = 0.79; *p* = 0.58. Ctr: control; Cit: citalopram; etx: etifoxine; D: vitamin D; AA: ascorbic acid; SEM, standard error of the mean.

**Figure 3 ijms-25-01960-f003:**
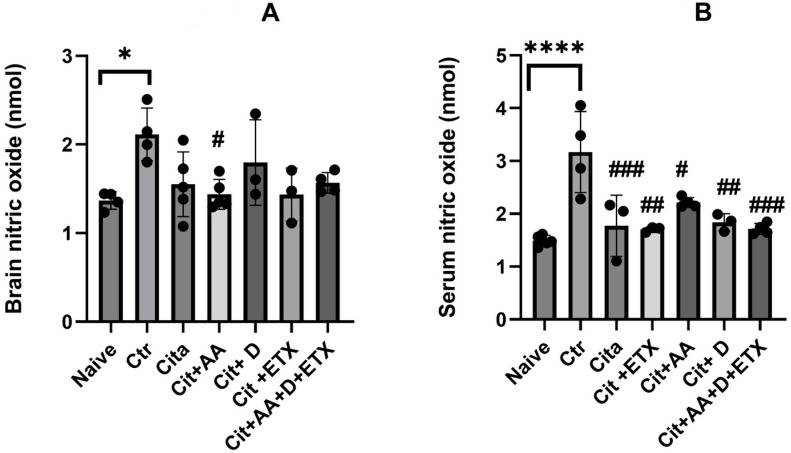
(**A**) Brain nitric oxide expression among the different groups ANOVA followed by Tukey’s post hoc analysis. Values are expressed as the mean SEM (ANOVA followed by Tukey’s test). F(6, 21) = 3.47; *p* = 0.01. * *p <* 0.05, # *p <* 0.05 versus Ctr. Ctr: control; Cit: citalopram; Etx: etifoxine; D: vitamin D; AA: ascorbic acid; NO: nitric oxide; SEM, standard error of the mean. (**B**) Serum nitric oxide expression among the different groups ANOVA followed by Tukey’s post hoc analysis. Values are expressed as the mean SEM (ANOVA followed by Tukey’s test). F(6, 19) = 9.75; *p <* 0.001. **** *p* <0.0001 versus naive, # *p <* 0.05, ## *p* <0.001, ### *p <* 0.0001 versus control. Ctr: control; Cit: citalopram; Etx: etifoxine; D: vitamin D; AA: ascorbic acid NO: nitric oxide; SEM: standard error of the mean.

## Data Availability

The data are available from the corresponding author.

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
