# Peer review of "Exploring the Roles of Vitamins C and D and Etifoxine in Combination with Citalopram in Depression/Anxiety Model: A Focus on ICAM-1, SIRT1 and Nitric Oxide"

_ijms, 2024, doi:10.3390/ijms25041960_

Round 1
Reviewer 1 Report
Comments and Suggestions for Authors
There is growing evidence that neuroinflammation is involved in depressive disorder.
The soluble intercellular adhesion molecule-1 65 (sICAM-1) in serum is considered a peripheral inflammatory marker associated with several inflammatory states. Since the blood levels of sICAM-1 were found to be increased in patients with depression the authors investigated in a mouse model the potential augmentation of citalopram by ascorbic acid (AA), Vitamin D (D), and etifoxine (Etx), which alleviates anxiety symptoms and related the antidepressant effect to brain and serum ICAM-1, the antioxidative and anti-inflammatory SIRT1, and NO.
The different combinations clearly showed positive effects.
Unfortunately, as indicated below, except Vit. C, the doses applied are much higher than those used in human therapy and most likely will induce severe side effects in man. With this the outcome of the study is of no human relevance and does not warrant publication.
Vit C 10 mg/kg = 700 mg per 70 kg, therapeutic daily dose 1000 mg and more
Etifoxin 50 mg/kg = 3,500 mg per 70 kg, therapeutic dose 150-200 mg/day, higher doses induce headache and dizziness
Citalopram 10 mg/kg = 700 mg per 70 kg, therapeutic doses 20-40 mg
Vit. D 1200 IUPer kg per week = 84,000 IU per week per 70 kg, therapeutic dose 28.000 IU per week.
Comments on the Quality of English LanguageDoses applied are beyond tehrapeutic doses and will have serious side effects.
Author Response
Kindly find the attached file below

Reviewer 2 Report
Comments and Suggestions for Authors
Gammoh and colleagues explored the roles of vitamins C and D, and etifoxine in combination with citalopram in depression/anxiety model. They focused their attention on ICAM-1, SIRT1 and Nitric Oxide. The Authors concluded that ICAM-1, Sirt1, and NO are implicated in depression and are modulated by antidepressants. Despite these findings would be of general interest to this field of research, the work has serious flaws that need to be addressed.
Major points
- The statistical analysis is incomplete. The Authors must show F and P values of the main factors of ANOVA analysis. Moreover, it is not clear the threshold of significance. What does “Significance was established at p 0.05” mean?
- The manuscript in many parts (mostly the results) is not easily readable. The description of the results is confused. This is a serious flaw. I suggest an extensive editing of English language.
- The Authors inappropriately used terminology for clinical studies. For example they wrote in the abstract “brain NO increased significantly (p 0.05) in depression and anxiety”. Mice exhibit anxiety-like or depressive-live behaviors. They do not develop depression or anxiety, which are human states.
- Line 187: (6.751.93) (24.014.0)…ecc… What do these numbers mean?
- The Authors must better discuss the effects of antidepressants on oxidative stress and NO signaling. The Authors must add these recent research papers: PMID: 35002742; PMID: 37972712 and others).
- The figures are inappropriately showed. The graphs 1A, 1B… must belong to a unique figure with a unique legend. The Authors must check all the figures.
Minor points
- There are several typos throughout the manuscript. For example: asserotoni, behavior-like behavior, Foating (figure 1A)..ecc..
- The Authors should avoid colloquial sentences.
- The sex of the animals is not indicated.
- There are many statements without references throughout the manuscript.
Comments on the Quality of English LanguageExtensive editing of English
Author Response
Kindly find the responses in the document provided

Reviewer 3 Report
Comments and Suggestions for Authors
Dear Authors,
I have a few comments on the research paper, which is entitled "Exploring the roles of vitamins C and D and etifoxine in combination with citalopram in the depression/anxiety model: A focus on ICAM-1, SIRT1, and Nitric Oxide."
1. In the title, there is an English correction, which is provided above.
2. The experimental design was not properly shown (n =?).
3. In the results section, please provide the graphs in the form of bar diagrams with dots.
4. The western blots do not match the original blots that you provided.
5. A single well blot for a group is not recommended. Please provide at least two in a group.
Author Response
Dear Reviewer, kindly find the responses in the document provided

Round 2
Reviewer 2 Report
Comments and Suggestions for Authors
The Authors have addressed all the issues I raised.
Comments on the Quality of English LanguageMinor editing